

# The role of antecedent soil moisture conditions on rainfall-triggered shallow landslides

Maurizio Lazzari[1], Marco Piccarreta[2], Salvatore  Manfreda[2]

[1] CNR-IBAM,  C/da S. Loja, Zona Industriale 85050 Tito Scalo (PZ) - Italy
[2] DICEM, Università degli Studi della Basilicata, Viale dell'Ateneo Lucano 10, 85100 Potenza, Italy

*Correspondence to*: Marco Piccarreta (marco.piccarreta@istruzione.it )

**Abstract.** Rainfall-triggered shallow landslides have caused losses of human life and millions of euros in damage to property in all parts of the world. The need to prevent such phenomena combined with the difficulty to describe the geo-physical processes over large scales led to the adoption of empirical rainfall thresholds derived from the observed relationship between rainfall intensity/duration and landslide occurrence. These thresholds are generally obtained neglecting the role of the antecedent moisture conditions that should be taken into consideration. In the present manuscript, we explored the role of antecedent soil moisture on the critical rainfall intensity–duration thresholds highlighting its critical impact. Therefore, traditional approaches that neglect such parameter may have a limited value in the early-warning systems. This study was carried out using a record of 326 landslides occurred in the last 18 years in the Basilicata region (southern Italy). Besides the ordinary data (i.e. rainstorm intensity and duration), we also derived the antecedent moisture conditions using a parsimonious hydrological model.

Key words: landslides, soil saturation, geomorphology, hydrogeological risk, Basilicata

## 1 Introduction

Rainfall-induced shallow landslides constitute an important issue of scientific and societal interest, causing billions of euros in damages and thousands of deaths each year (Hong et al., 2006). A large number of studies investigated the functional relationship between rainfall events and landslides. One of the main results is the definition of empirical rainfall thresholds, such as intensity-duration, event-duration and event-intensity thresholds, thresholds based on the total event rainfall, thresholds that consider the event rainfall or snowmelt amounts to predict the occurrence of such events (Guzzetti et al., 2008 and reference therein; Lazzari et al., 2013; Segoni et al., 2018). However, this approach is affected by a large number of false positives, and, moreover by a limited physical process understanding (Bogaard and Greco, 2018). All these thresholds indeed use as just physical parameter the precipitation, neglecting the primary role of other parameters like evapotranspiration, soil moisture, rainfall infiltration, soil porosity, permeability, ecc.
In order to consider predisposing hydrological factor on empirical threshold calculation, recent studies have focused on the role of the antecedent water content in landslides dynamics (Brocca et al., 2012; Coe, 2012; Crozier, 1999; Glade et al.,



2000; Godt et al., 2006, 2009; Ponziani et al., 2012). These approaches have found a strong relation between the rainfall values triggering landslides and the initial soil moisture contributing to improve the predictive accuracy of empirical threshold. Those results have also stimulated a critical revision of the I/D thresholds in the last few years (Bogaard and Greco, 2018; Mirus et al., 2018a and 2018b; Valenzuela et al., 2018). In particular, Bogaard and Greco (2018) introduced the

5 cause-trigger concept for defining hydro-meteorological regional landslide thresholds. This approach combines the antecedent factors that cause hillslopes to be predisposed to failure and the actual trigger associated with landslide initiation. Starting from this new perspective, we aim to contribute to this discussion by analysing the role of antecedent soil moisture conditions on rainfall-triggered shallow landslides. Much of the quoted paper have not shown quantitatively how the antecedent moisture conditions influence the rainfall thresholds leaving different question still open. How much and how can

the initial saturation degree affect the I/D relationships?

In this paper, we try to answer to those questions exploiting a record of 326 landslide occurrence in the Basilicata region (southern Italy, Fig.1), from 2001 to March 2018. For each of the landslide event identified we derived the rainfall event characteristics and also the antecedent soil moisture conditions, using a parsimonious hydrological model applied at the regional scale over a temporal window of 18 years.

## 2. Data and methods

Basilicata is a Southern Italy region highly vulnerable to hydrogeological risk due to landscape typologies (47% mountains,

45% hillocks and 8% plains). The precipitation regime is Mediterranean, with distinct dry and wet seasons (Piccarreta et al., 2013). Higher precipitation totals occur during the last autumn–winter period when landslides and floods normally take place (more than 70 %). On the basis of a detailed bibliographical research (Lazzari, 2011; Lazzari and Gioia, 2015; Lazzari et al., 2018) including national and local newspapers and journals, Internet blogs, and the scientific and technical literature we have collected 326 shallow landslide events from January 2001 to March 2018. The territory is covered by a near real-time

hydrometeorological network, that are uniformly distributed on the territory (1 station every 80 km2) and characterized by uninterrupted and high-quality data (recording time interval from 1 to 60 min).

Although antecedent soil moisture can be obtained by in-situ measurements, regional scale measurements are time consuming and expensive. In recent times, more information is available from satellite data, but they also are too course in order to provide local estimates of soil water content on a specific landslide (Ray and Jacobs, 2007; Brocca et al., 2012).

Thus, we adopted the hydrological model AD2 (Manfreda et al., 2005; 2014; 2018) to derive the antecedent soil moisture before each rainfall event identified in our database. Simulation was carried out using about one year of rainfall data and temperature in order to reach a reliable estimate of the relative soil water content before the considered event. AD2 provides a hydrological prediction taking into consideration the following hydrological components: infiltration, surface runoff, sub-



surface runoff, deep percolation, and evapotranspiration. Soil water balance soil the is described by the following equation (Farmer et al., 2000):

$$S_{t+\Delta t} = S_t + I_t - R_{out,t} - L_t - E_t \tag{1}$$

where $S_t$ is the basin soil water content at the generic instant of time t, which represents a key variable of the model influencing runoff production, leakage and evapotranspiration; $I_t$ is the infiltration; $R_{out,t}$ is the sub-surface runoff production; $L_t$ is the leakage to the groundwater; and $E_t$ is the actual evapotranspiration.

The model adopts a conceptual schematization with physically consistent parameters that allowed to assign model parameters based on soil texture. This allowed to derive retrospective relative saturation values consistent with the local dynamics of the soil. This approach is very simple and can be easily replicated elsewhere after a simple calibration against the local soil moisture and landslide datasets. It must be stated that obtained values can be affected by several errors due to model structure, parametrization, and climatic data, but at the present such an approach provides a realistic description of the expected relative saturation providing a synthesis of the state of the system according to the available information on soil texture, antecedent rainfall, and evolution of temperatures.

To determine rainfall thresholds for possible shallow landslide occurrence, we adopted the Frequentist method (Brunetti et al., 2010). The threshold curve is assumed to be a power law:

$$I = \alpha D^{-\beta} \tag{2}$$

where I is the rainfall mean intensity (mm h$^{-1}$), D is the rainfall event duration (h), α is the intercept and β defines the slope of the power law function. Empirical data were log-transformed to calculate the best-fit line by means of a linear equation $\log(I) = \log(\alpha) - \beta \log(D)$, equivalent to that described above.

Following the methodologies also adopted in previous studies (Godt et al., 2006; Ponziani et al., 2012; Valenzuela et al., 2018), we have directly correlated the rainfall event associated to each landslide event and identified the corresponding degree of saturation of the soil at the starting time of each landslide event. Including this additional information in the database, it was possible to explore the role of antecedent soil moisture on landslides induced by rainfall events.

## 3. Results and discussion

The hydrological model AD2 was adopted to describe the evolution of soil moisture over the entire Basilicata region using available information about physical characteristics of the soils and also available meteorological data extracted from the



regional network (Fig. 1) for the period 1/1/2001 to 28/03/2018. This allowed to derive the evolution of the degree of saturation of the soil in each location, where a landslide event was recorded.

An illustrative example of the evolution of the main hydrological variables is given in Figure 2, which provides the temporal evolution of the rainfall and relative saturation of soil of three different sites (Lauria, Vietri di Potenza and Pisticci; see

figure 1) characterized by different lithological conditions during the period from 1/1/2009 to 31/12/2015. It can be noticed that most of the different landslide events (reported in the graph with a dark star) occurred after significant rainfalls amounts occurring in conditions of relatively high or moderate soil saturation degree. The same rainfall amounts, occurred in conditions of low antecedent soil moisture content, did not produced landslides. As already found by numerous previous studies (Brocca et al., 2012; Coe, 2012; Crozier, 1999; Glade et al., 2000; Godt et al., 2006, 2009; Mirus et al., 2018a and

2018b; Ponziani et al., 2012; Valenzuela et al., 2018), several rainfall events of significant amount did not produce any shallow movements. This preliminary plot show that there is an interplay between the antecedent soil moisture conditions and the amounts and the duration of the triggering rainfall events.

By following the trigger-cause concept of Bogart and Greco (2018), to account for the role of the antecedent wetness condition on rainfall thresholds, for each landslide event the rainfall intensity/duration has been plotted versus the simulated

antecedent soil saturation (Fig. 3). For each rainfall-induced shallow landslide the relative degree of saturation has been referred to the start day of the triggering rainy event. Similarly to previous studies (Crozier 1999; Baum and Godt 2009, Glade et al., 2000; Ponziani et al, 2012), a linear decreasing trend between the rainfall thresholds and the initial soil moisture conditions has been found. The correlation between increasing levels of soil saturation and decrease of maximum rainfall is particularly clear for rainfall durations lower than 48 h. For longer rainfall durations (up to 48 h) the correlation becomes

weaker. This is probably due to the fact that over longer rainfall events the total amount of water entering into the soil increases and also the influence of the initial as a consequence decreases.

A clear picture of the dependence between rainfall intensity thresholds and the antecedent soil moisture is given in Fig. 4, where the rainfall intensity values of the 326 investigated events are associated the simulated soil saturation degree using a colour scale (from blue to yellow starting from lower to higher values of degree of saturation). It clearly emerges that higher

values of rainfall intensity correspond to lower values of soil saturation and vice-versa.

To evaluate the role of soil degree saturation on the regional rainfall threshold, the dataset of triggering rainfall events has been subdivided in different groups based on the relative value of antecedent relative saturation. We tested different relative saturation thresholds and obtained that significant difference could be observed using a reference value of relative soil saturation equal to 0.70. Therefore, dataset was divided in one group of events occurring under middle-low antecedent soil

moisture conditions (relative soil saturation lower than 0.70), and one under medium/high antecedent soil moisture conditions (relative soil saturation higher than 0.70). In this way, it was possible to derive critical rainfall threshold functions conditional on the antecedent soil moisture conditions. The obtained functions synthetize the impact of relative saturation of soil on the triggering of landslide events. In order to quantitatively evaluate the differences between the two different thresholds, in Table 1 we have reported the critical rainfall thresholds estimated for durations ranging from 1 to 120 hours




for three levels of probability equal to 0.6, 0.75 and 0.9. The table illustrates the impact of such choice on the relative values assumed by the estimated rainfall threshold expressed in terms of cumulated rainfall.

## 4. Conclusions

The effects of antecedent soil moisture conditions on rainfall I/D thresholds triggering shallow landslides is explored using a dataset build for a region of southern Italy. We found that antecedent soil saturation degree conditions play a crucial role on landslide triggering, which may support the improvement of forecast systems. Combining rainfall events data with the antecedent soil moisture conditions, it was possible to derive I/D relationships able to better discriminate the triggering

conditions for landslides. Two distinct soil degree saturation values ($< 0.7$ and $> 0.7$) were identified in order to distinguish different class of events. Such soil moisture conditions leaded to two distinguished population of events that identified significantly different rainfall threshold functions. Such differences may be linked to the induced difference in surface hydrology that may be due to the triggering of subsurface redistribution when soil moisture exceeds field capacity.

In many regions, landslide warning systems are based on simple rainfall thresholds. In our opinion, the calculation of soil

saturation should be the first step for an effective prediction of real-time landslides risk decreasing the uncertainties tied to the application of the rainfall thresholds only.

The I/D rainfall thresholds are not really useful for an efficient warning system, because they usually lead to a large increase of false alarms or, conversely, in an underestimation of real risk. Therefore, the use of this additional physical information may further support the reliability of conditional threshold functions considering antecedent soil moisture conditions. More

extensive studies have to be done to further explore rule of additional factors such as geology and pedology of the site.

Rainfall data and soil degree saturation from a Southern Italy region are used to explore the effects of antecedent soil moisture conditions on rainfall I/D thresholds triggering shallow landslides. We found that previous soil degree saturation conditions play a crucial role on landslide triggering, much more than the same rainfalls. Soil saturation deeply affects the rainfall intensity–duration threshold, showing how this last approach is not really useful alone in prevent rainfall-induced

landslides triggering. Two distinct soil degree saturation values ($< 0.7$ and $> 0.7$) clearly affects the amount and the duration of rainfalls triggering shallow landslides, producing deep variations in the I/D relationships. Although some conclusions may seem obvious, they are not. In many regions of our planet, landslide warning systems are based on simple rainfall thresholds. In our opinion, the calculation of soil saturation should be the first step for an effective study on the real-time landslides risk assessment and define a more reliable scenario for the regional territory, decreasing the uncertainties tied to the application

of the rainfall thresholds only. The rainfall I/D threshold are not really useful for an effective warning systems, because they usually consists in a large increase of false alarms or, conversely, in an underestimation of real risk. If they are coupled with antecedent soil moisture condition their use acquires more and more significance. More extensive studies have to be done to fully explore the regional control of soil degree saturation, in terms of geological and pedological conditions.





**Acknowledgements**

This work was carried out within a scientific agreement between the Civil Protection Department of Basilicata, the Interuniversity Consortium for Hydrology (CINID), and the University of Basilicata to the start-up the Basilicata Hydrologic
Risk Center. We would like to thank reviewers for their insightful comments on the paper, as these comments led us to an improvement of the work.

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





**Figure Captions**

Figure 1 – Geographical distribution of the weather stations for the study area and the location of the three different sites
5 (Lauria, Vietri and Pisticci), characterized by different lithologies, used to illustrate the procedure adopted herein.

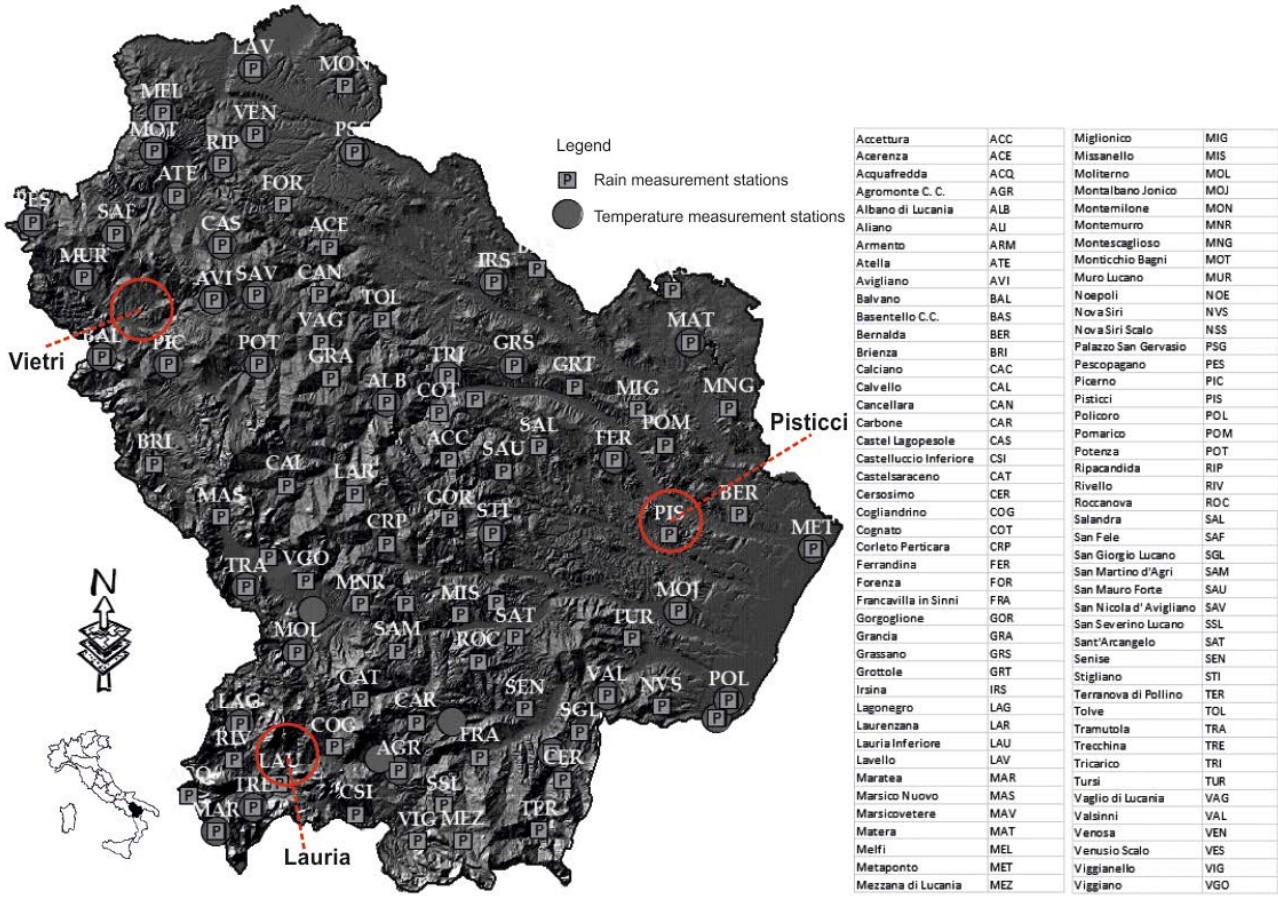




Figure 2 – Daily rainfall (blue) vs simulated daily soil degree saturation (red) at (a) Lauria, (b) Vietri and (c) Pisticci from 1/1/2009 to 31/12/2016. Dark stars represent the data of occurrence of shallow landslide events in the monitored areas.

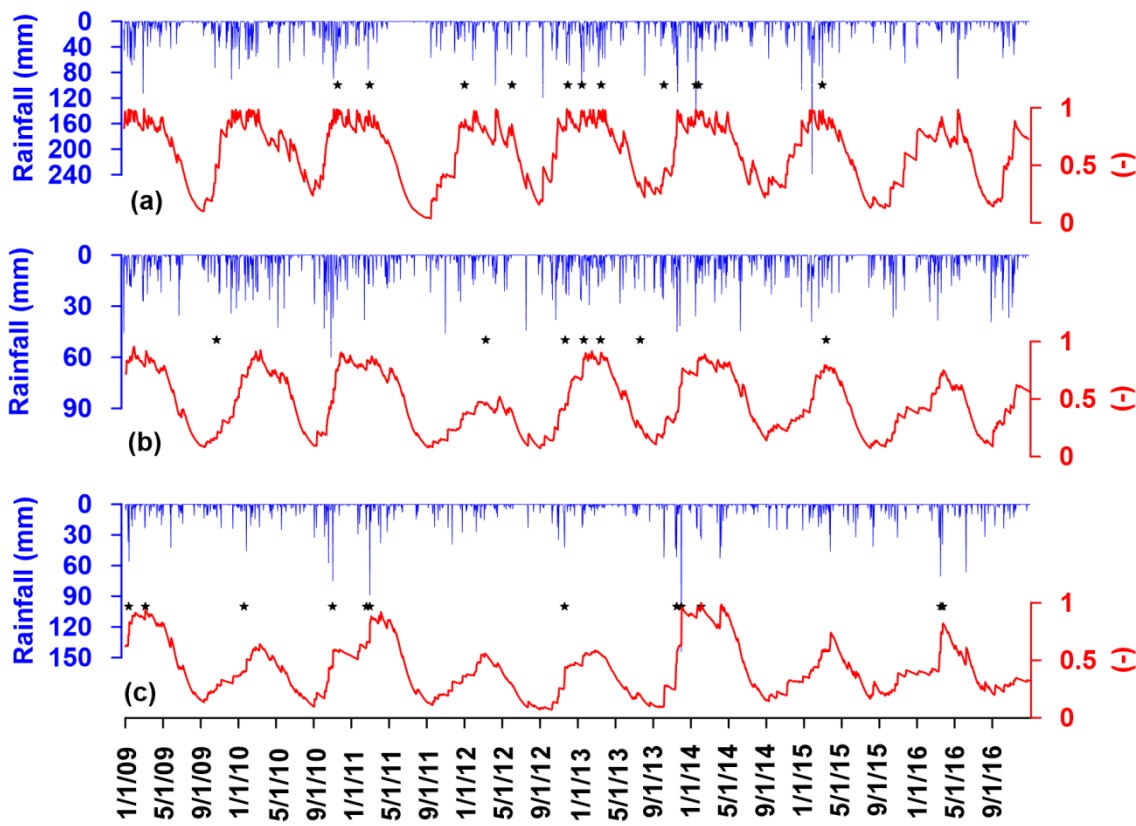




Figure 3. Rainfall intensity/duration versus the simulated initial degree of saturation for the 326 landslide events in Basilicata region (Southern Italy) from 2001 to 2018

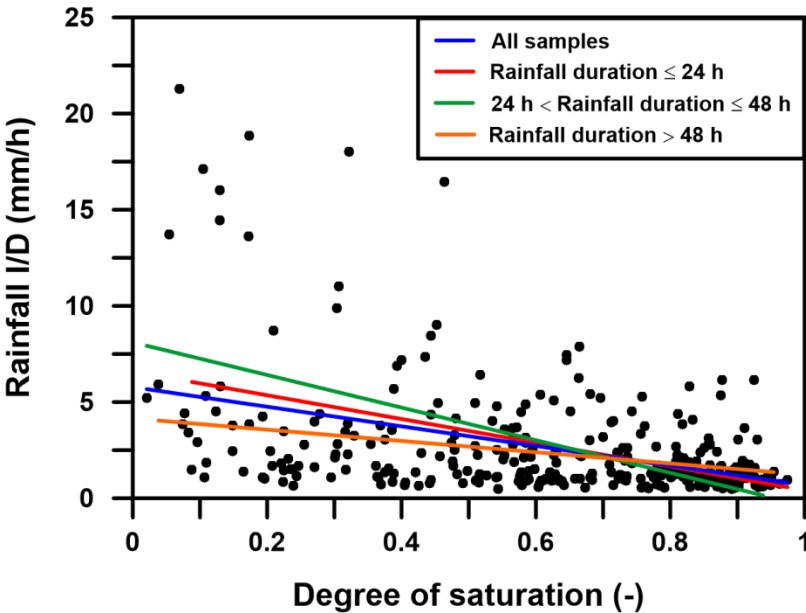

Figure 4. Rainfall intensity versus the simulated antecedent degree of saturation for the 323 landslide events in Basilicata region (Southern Italy) from 2001 to 2018. Full line represents the critical rainfall threshold obtained from observations with soil degree saturation lower than 0.70, while dashed line represents the critical rainfall threshold obtained using observations with soil degree saturation higher than 0.70.

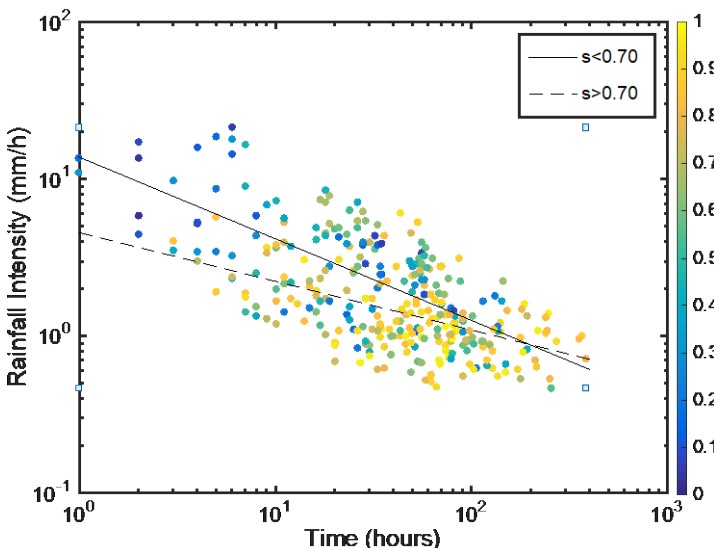



Table 1. Critical rainfall thresholds estimated for durations ranging from 1 to 24 hours for three levels of probability equal to 0.6, 0.75 and 0.9.

| Degree of saturation < 0.70 | | | | Degree of saturation > 0.70 | | | |
|---|---|---|---|---|---|---|---|
| Time span (hours) | H1 | H2 | H3 | Time span (hours) | H1 | H2 | H3 |
| 1 | 13,05 | 17,00 | 24,89 | 1 | 6,59 | 8,15 | 11,08 |
| 2 | 18,80 | 24,50 | 35,87 | 2 | 10,22 | 12,64 | 17,17 |
| 3 | 23,28 | 30,33 | 44,41 | 3 | 13,20 | 16,33 | 22,19 |
| 4 | 27,10 | 35,30 | 51,68 | 4 | 15,84 | 19,59 | 26,62 |
| 5 | 30,48 | 39,71 | 58,13 | 5 | 18,24 | 22,56 | 30,66 |
| 6 | 33,55 | 43,71 | 64,00 | 6 | 20,47 | 25,32 | 34,40 |
| 7 | 36,39 | 47,41 | 69,42 | 7 | 22,56 | 27,91 | 37,93 |
| 8 | 39,04 | 50,87 | 74,48 | 8 | 24,55 | 30,37 | 41,27 |
| 9 | 41,55 | 54,12 | 79,25 | 9 | 26,45 | 32,72 | 44,46 |
| 10 | 43,92 | 57,22 | 83,77 | 10 | 28,27 | 34,98 | 47,53 |
| 11 | 46,18 | 60,16 | 88,09 | 11 | 30,03 | 37,15 | 50,48 |
| 12 | 48,35 | 62,99 | 92,22 | 12 | 31,73 | 39,25 | 53,34 |
| 18 | 59,87 | 77,99 | 114,20 | 18 | 41,01 | 50,73 | 68,93 |
| 24 | 69,67 | 90,77 | 132,90 | 24 | 49,19 | 60,85 | 82,69 |
| 36 | 86,27 | 112,39 | 164,56 | 36 | 63,58 | 78,65 | 106,87 |
| 48 | 100,39 | 130,79 | 191,50 | 48 | 76,26 | 94,34 | 128,20 |
| 60 | 112,93 | 147,12 | 215,41 | 60 | 87,83 | 108,64 | 147,63 |
| 72 | 124,32 | 161,96 | 237,13 | 72 | 98,56 | 121,93 | 165,68 |
| 84 | 134,84 | 175,67 | 257,21 | 84 | 108,66 | 134,41 | 182,65 |
| 96 | 144,67 | 188,48 | 275,96 | 96 | 118,23 | 146,26 | 198,75 |
| 108 | 153,94 | 200,55 | 293,64 | 108 | 127,38 | 157,57 | 214,12 |
| 120 | 162,73 | 212,00 | 310,40 | 120 | 136,16 | 168,43 | 228,87 |

