# Peer review of "The role of antecedent soil moisture conditions on rainfall-triggered shallow landslides"

_Natural Hazards and Earth System Sciences, 2018_

## Referee Comment (RC1) · Anonymous Referee #1 · 28 Dec 2018

GENERAL COMMENT

The manuscript prepared by Lazzari and co-workers addresses an important scientific and technical question, i.e. the prediction of rainfall induced landslides with the use of antecedent soil moisture conditions obtained from a hydrological model. The topic fits the scope of NHESS and is interesting for the readers. The work is based on a dataset of landslides occurred in a southern Italian region and adopts a hydrological model already presented in the literature. The work follows a relatively new research line in the definition of empirical/hydrological thresholds for the prediction of rainfall-induced landslides. The paper is well-structured, in a sufficient English language, and follows the IMRAD structure.

Despite the good intentions, I believe that the work is missing the goal, for a number

of reasons. I have found several lacks and drawbacks in the whole manuscript, which does not allow for its acceptance in the present form. Moreover, the conclusions are not supported by the obtained results. It seems to me that the Authors want to express strong criticisms to the "classical" methods used to calculate empirical rainfall thresholds for operative landslide prediction, but without operating the long, detailed and rigorous process needed for their adoption and definition.

My opinion is that the paper needs a strong improvement before being reconsidered for discussion and eventually publication. For this reason, I believe that now it should undergo through major revisions.

In the following, I list some specific comments and a list of technical corrections. All these comments should be carefully addressed and all the corrections should be done before the paper can be reviewed again.

SPECIFIC COMMENTS

First, the Authors start from a consideration about the "classical" approach for the definition of empirical rainfall thresholds. They state that it "is affected by a large number of false positives" (page 1, line 31 and elsewhere in the conclusions). This is not generally true. The number of false positives, as the number of false negative, is strongly related to the values (let me say, the height) of the rainfall thresholds. High thresholds produce many false negatives and few true positives, while low thresholds result in several false positives and limited true negatives. Therefore, the number of false positives is related to the values adopted for the definition of the thresholds. The definition of high or low thresholds is due to several issues.

There are classical thresholds that are low, therefore resulting in several false alarms, and other thresholds that are high, producing less false alarms. This should be acknowledged, also referring to some papers dealing with threshold validation. I suggest a huge literature review on this topic.

Second, regarding the landslides dataset, the Authors do not describe anything about the records. A description of several details would be very useful for understanding the quality of the used dataset. As an example, the description could be focused on: (i) the landslides types, (ii) the annual and monthly distributions of the landslides, (iii) their geographical distribution, (iv) the temporal (is the time or the day of occurrence known for all the records?) and spatial (are the coordinates of the landslides known?) accuracy in the identification of the landslides.

Moreover, always regarding the landslide dataset, the Authors often refer to "landslide events". What does it mean? How these landslide events were defined?

Third, moving to rainfall, nothing is said regarding the reconstruction of the triggering rainfall events. Nothing is said about the selection of the rain gauges necessary to associate the rainfall data to the landslide trigger. Nothing is said about the separation of rainfall events and about their association to the landslides (even if the Authors cite in several places these "rainfall events"). This is a great drawback that should be solved and discussed.

The same goes for the reconstruction of the soil moisture conditions associated to the landslides. In the section describing the method, nothing is said about it. On the other hand, in the "result" section, Authors state that the degree of saturation is related to the start day of the triggering rainfall events (whose definition is murky). This is another key point that should be better presented in the "method" section.

Fourth, regarding the AD2 model, nothing is said about the all the variables and parameters reported in equation 1. As an example, how the evapotranspiration is calculated/estimated? What about the infiltration? This is another point that should be better discussed.

Fifth, since nothing is reported about the rainfall data and rainfall events, it seems – looking at figure 2 – that daily rainfall measurements are used. This is not feasible when working with shallow landslides (as reported by in Section 2). On the other hand,

Authors state that detailed measurements are available (page 2, line 26). This issue should be clarified and discussed.

Sixth, in the results, the Authors state that they were able to "derive critical rainfall threshold functions" (Sections 3 and 4). However, the only threshold function visible in the manuscript is the equation 2, reporting the general form of an intensity-duration threshold. Then, in Table 1, one can read only a list of numbers (named "rainfall thresholds" by the Authors) for which not even the unit of measurement is provided. Sincerely, I do not understand how this list of values (of what? This is not clear) could be considered rainfall threshold functions. This is another key point that needs to be improved and discussed.

Finally, the "Conclusions" section is written in a way worse than the rest of the text, with several repetitions (see e.g, page 5 lines 6-7 and lines 21-22; line 14 and line 27; lines 17-19 and lines 30-32). It should be completely rewritten with more accuracy. Moreover, in my opinion, is full of wishful thinking, not supported by the obtained results. The Authors state that "the calculation of soil saturation should be the first step for an effective prediction of real-time landslides risk decreasing the uncertainties tied to the application of the rainfall thresholds only". This is a strong statement (and I could also agree with this aim) but it is not supported by any other statement, and not even by the obtained results. How the proposed method can be implemented in an "operative" landslide warning system? How the uncertainties can be reduced? Please note that the uncertainties related to the presented method are not even evaluated in the paper. Therefore, I cannot understand how this can reduce the uncertainties in the whole process of landslide prediction.

I have some other more specific comments, which are listed below.

Page 1, line 19: I would suggest adding a final statement in the abstract to describe the results and main findings of the work.

Page 1, line 24: there are some papers describing global datasets of landslides in-

duced by rainfall, even with fatal consequences, that should be mentioned.

Page 1, line 26: please use definitions more precise than "intensity-duration, event-duration and event-intensity thresholds". As an example, "intensity-duration" could be "rainfall mean intensity-rainfall duration". Furthermore, does "event-duration" mean "cumulated event rainfall-rainfall duration"?

Page 1, line 28: I would remove the self-citation here, since the mentioned paper by Lazzari et alii is not a review paper like the other two cited works. The reference to the work by Lazzari et alii could be moved elsewhere.

Page 1, line 30: please note the classical thresholds are based on "rainfall" measurement, not "precipitation" (e.g., usually snow or hail are not considered).

Page 2, line 12: please define "landslide event" and "rainfall event".

Page 2, lines 24-26: I would move this description of the hydro-meteorological network at line 22, after the description of the precipitation regime and before the description of the landslide dataset.

Page 3, lines 26-27 and page 4, lines 1-2: this part should be moved to the method section.

Page 4, lines 8-11: this sentence is quite vague. I see the point, but I believe that could be improved.

Page 4, lines 18-21: here the Authors state that the correlations found are "clear". However, I can't grab it. From the figure I can just see some best fit lines (I guess) without any correlation coefficient useful to quantitative assess the goodness of the correlation. This point should be better discussed.

Page 4, lines 30-31 and page 5, line 10: what about for relative soil saturation equal to 0.7?

Page 5, line 1: here the Authors refer about three levels of probability equal to 0.6,

0.75, and 0.9. I do not understand: (i) how the probability is defined and to what is related. (ii) how the three levels are defined and why these values are chosen.

Page 5, lines 6-33: please read and re-write accurately the whole paragraph, since it is full of repetitions of sentences, in some cases exactly alike. Please add the main findings of the work and how they can be used quantitatively for operative landslide prediction.

FIGURES AND TABLES

Figure 1.

The figure is very dark and very difficult to read. I would suggest using colours or a brighter DTM.

Moreover, I would suggest deleting the table with the names of the stations and the labels of the stations in the map. They are not useful to the discussion. Just leave the indications of the three sites. Please add scale bar and coordinates.

On the other hand, a similar figure with the distribution of the 326 landslides would be useful.

Figure 2.

I would suggest using the same maximum values on the y-axes for the three panels.

Moreover, I would suggest using the labelling "d/m/y" for the x-axis and adding the final date.

As already mentioned, I cannot understand the use of daily rainfall, in particular given that more detailed measurements are available.

Figure 3.

The caption of the figure says "Rainfall intensity/duration" and the label of y-axes "Rainfall I/D (mm/h)". The unit of measurement is related to an intensity. So, is it just rainfall

intensity? Please explain.

Please correct "landslide events" in the caption.

Moreover, are the lines reported in the figure the best fit lines of the point clouds? If yes, please add the equations and the values of the correlation coefficients. If not, please explain what they are.

Finally, for an optimal representation of the different groups of rainfall durations, I would suggest to use different colours (the same of the lines) for the points pertaining the each of the three groups.

Figure 4.

What about for relative soil saturation equal to 0.7?

Does the x-axis represent the duration? In any case, use only "h" instead of "hours". Please add "s" in the caption. There are two squares in the right part of the figure. What they are?

Moreover, are the lines reported in the figure the best fit lines of the point clouds? If yes, please add the equations and the values of the correlation coefficients. If not, please explain what they are.

I appreciate the use of the colour scale for indicating the different values of soil saturation; however, I suggest adding the description in the caption and not only in the text. Finally, for an optimal representation of the figure, I would suggest to use different symbols (e.g. circles and squares) to represent points with soil saturation higher or lower than 0.7 (but please consider also values equal to 0.7).

Table 1.

Please explain what the numbers included in the table represent. Are they values of rainfall mean intensity? Are they values of cumulated event rainfall? Please explain and add the description in the caption.

What are H1, H2, and H3? Please explain.

Please use only "h" instead of "hours". Please use point as decimal separator. Please justify the use of two decimal places.

TECHNICAL CORRECTIONS

Page 1, line 13: I would replace "rainfall intensity/duration" with "rainfall mean intensity and duration".

Page 1, line 15: please avoid repetition of "critical" in the line. Please delete "the" before "critical rainfall thresholds".

Page 2, line 3: please define "I/D".

Page 2, line 4: please replace "Mirus et al. 2018a and 2018 b" with "Mirus et al. 2018a, b". Also at page 4, lines 9-10.

Page 2, line 5: I would replace "regional landslide thresholds" with "regional thresholds for landslide prediction".

Page 2, line 11: please replace "record" with "dataset".

Page 2, line 12: please add "January" before 2001.

Page 2, line 19: please use "types" instead of "typologies".

Page 2, line 28: please correct "course".

Page 3, line 2: please note that "Farmer et al. 2000" is reported with the year 2003 in the reference list. Please check.

Page 3, line 3 and line 6: I would suggest using the most common acronym "ET" to refer to the evapotranspiration, in order to avoid confusion with E – evaporation.

Page 3, lines 4-6: please check the subscripts of the variables.

Page 3, line 20: I would suggest using "methods" instead of "methodologies".

Page 4, lines 3-5: please correct the punctuation in the sentence.

Page 4, line 5: please note that in the text the date is "31/12/2015" while in the Figure 2 (and caption) is "31/12/2016".

Page 4, line2 6-7: please avoid repetition of "occurred/occurring".

Page 4, line 11: please correct "show".

Page 4, line 13: please correct "Bogart" into "Bogaard".

Page 4, line 16: I would suggest using "rainfall events" instead of "rainy events".

Page 4, line 23: is the "rainfall intensity" the "mean rainfall intensity"?

Page 4, line 23: please correct "are associated the simulated..."

Page 4, line 26: please replace "threshold" with "thresholds".

Page 5, line 6: please change "thresholds triggering shallow landslides" into "thresholds for the triggering of shallow landslides".

Page 5, line 7: please correct "build".

Page 5, line 11: please replace "class" into "classes".

Page 6: please add DOI where available.

Page 7, lines 13-14: I've found this paper with a different title.

---

## Author Comment (AC1) · 21 Jan 2019

Reply to Reviewer 1

We would like to express our gratitude to the reviewer for his hard work and constructive criticism. Many observations are worthy of consideration and will allow us to improve and rebalance different parts of the work. At the same time, we must point out that being this a short communication, we had to summarize many of our activities giving more emphasis to the outcomes of the research and neglecting some information.

Going into the details of the reviewer comments, we provide a point by point comments in the following.

1) Regarding to the literature, we have considered a large part of literature on rainfall

thresholds. Many of their results are reported in review work quoted in bibliography (Brunetti et al., 2010; Guzzetti et al., 2008; Segoni et al., 2018). Given the editorial rules on a short communication and having to give more space to data and discussions, we thought to summarize the introduction. However, we will certainly integrate the introduction with a more detailed bibliographic analysis, taking into account works on threshold validation.

2) We had to make a strong synthesis of the text to follow the editorial rules in terms of short communication. To overcome this problem we will redraw Figure 1 (as also suggested by the referee), inserting the geolocalizations of landslides and a graph with their monthly distribution. The information requested by the referee will be inserted in the text, in order to better explain the available data and their treatment methodology. Regarding to the climatic data, as explained in the text ( Page 3 line 18-20) the territory is covered by a near real-time hydrometeorological network, that are uniformly distributed on the territory (1 station every 80 km2) and characterized by uninterrupted and high-quality data (recording time interval from 1 to 60 min). We have used hourly data.

3) A database of rainfall events that have resulted in shallow landslides was compiled from 2001 to 2018. The landslide information was obtained through a detailed bibliographical research [Lazzari, 2011; Lazzari and Gioia, 2015; Lazzari et al., 2018] including national and local newspapers and journals, Internet blogs, and the scientific and technical literature. The collected information was organized in a catalogue listing 326 shallow landslide events from January 2001 to March 2018. For each rainfall event, the information collected and stored in the catalogue includes (see new Figure 1): the precise or approximate location of the area affected by the rainfall and the landslides; the precise or approximate time, date, or period of the failures; the rainfall conditions that resulted in slope failures, including the total event rainfall, the rainfall duration, the mean rainfall intensity, and the antecedent rainfall for different periods; the landslide type; a generic description of the main rock types. Hourly rainfall and temperature

is obtained from the raingauges of the Civil Protection. The rainfall duration D was determined measuring the time between the moment, or period, of initiation of the failure(s) (rainfall ending time) and the time when the rainfall event started (rainfall starting time). The rainfall ending time was taken to coincide with the time of the last rainfall measurement of the day when the landslide occurred. As suggested by Brunetti et al. (2010), for the identification of the starting time has been considered a minimum period without rain (a 2-day period without rainfall was selected for late spring and summer, May–September, and a 4-day period without rainfall was selected for the other seasons, October–April) to separate two rainfall events. Once the duration of the event was established, the corresponding rainfall mean intensity I (in mm h-1) was calculated dividing the cumulated (total) rainfall in the considered period (in mm) by the length of the rainfall period(in hours). Using this method, the rainfall mean intensity for the event was determined.

4) We agree with the reviewer and more details about the hydrological modelling and calibration will be given in the revised version of the manuscript.

5) In the work were actually calculated of rainfall thresholds as shown in figures 3 and 4. We neglected reporting the equations obtained. We have reported on Table 1 the obtained critical rainfall thresholds estimated for several durations to better highlight the different impact in using different threshold as a function of soil saturation estimates. However, we realize that these data do not provide an essential contribution to this discussion. Therefore, the table will be removed giving more emphasis to the rainfall thresholds conditional on the antecedent soil moisture conditions. Moreover, a second extended manuscript on the use of the proposed procedure is already in progress, where the methodology will also be validated.

6) Finally, we understand that some of the conclusion must be revised after a more robust validation of the proposed methodology. The aim of the present short communication is only to underline the strong control played by antecedent soil moisture condition on rainfall induced landslides. Therefore, the conclusion will be reorganized

removing any reference to warning systems. This will certainly be a second step of the research. All specific corrections will be made in the final review.

[Figure]

**Landslide monthly frequence (%)**

| Month | % |
|-------|---|
| Jan | 17% |
| Feb | 21% |
| Mar | 21% |
| Apr | 4% |
| May | 1% |
| Jun | 1% |
| Jul | 3% |
| Aug | 2% |
| Sep | 2% |
| Oct | 5% |
| Nov | 10% |
| Dec | 13% |

2184m a.s.l.

0m a.s.l.

0   10   20
km

**Legend**

**Landslide location**

○  Certain location within a radius of 1 km

●  Certain location within a radius of 5 km

▲  Climatic station

**Fig. 1.**

---

## Referee Comment (RC2) · Mirus (Referee) · 1 Feb 2019

This study addresses an important issue related to representing antecedent soil moisture conditions in thresholds for landslide warning systems. The manuscript has the potential to contribute to a growing body of literature to advance process understanding of landslide initiation and improve future development of hydro-meterological thresholds. The writing and figures are mostly clear, but some details are not obvious, and others are simply not provided. The manuscript was submitted as a brief communication, but I think it might need to be slightly longer to be fully informative. Although the study seems to be relevant and displays some promising results that could ultimately be of interest to readers of NHESS, the lack of rigor and important details is problematic. My primary concern when evaluating the technical merit of this work is the lack of details

available about the data, modeling, and analysis used to develop the thresholds and arrive at the conclusions. Therefore, the manuscript should undergo major revisions to address the following general question and issues before the work can be evaluated further.

1. Information is needed about the size of the field area and the actual variables measured.

2. Details are missing about the model equations, input/output, physical parameters, parameterization or calibration, and spatial and temporal resolution.

3. The simulated soil moisture does not seem to be compared to any observed soil moisture data, so the accuracy of the model output is highly uncertain.

4. It is unclear how the various antecedent saturations were quantitatively evaluated to select the significant value of 0.70.

5. It appears that the thresholds for >0.70 and <0.70 were optimized by identifying a best-fit line through the landslide data, rather than identifying a threshold that distinguishes between landslide and non-landslide events. This is highly unusual and needs to be justified.

6. Furthermore, there does not appear to be any quantitative analysis of the threshold performance for predicting landslide events, such as ROC analysis or other statistical metrics.

In addition to these general concerns, the following specific comments are worth noting in the revisions.

The abstract needs to focus more on the actual study and results. The motivation is important, but an informative abstract should include a clear description of the methods and state the primary contributions.

P1,L11. Preventing landslides seems like a nearly impossible goal and beyond the

**NHESSD**

[Figure]

scope of this study, but reducing the losses and impacts is more achievable through developing better landslide thresholds for early warning. That is the focus of this work.

P2,L1&4. The Mirus et al. 2018a,b references might be more appropriate to cite in line 1 as we actually developed new thresholds that improve predictive capabilities, though they do not explicitly consider rainfall I/D. Also consider adding Thomas et al., 2018, Geophysical Research Letters, doi:10.1029/2018GL079662, which uses a deterministic approach with infiltration simulations to identify rainfall-saturation thresholds.

P2,L8-9. Thomas et al., (2018) also does a nice job of quantifying the sensitivity of thresholds for different hydraulic and strength properties in relation to rainfall-saturation thresholds, though it doesn't deal with I/D directly. Godt et al., 2006 directly examines how antecedent moisture index affects the accuracy of an I/D threshold. It is not clear how/why your objectives should be distinguished from these prior advances.

P2,L9. Question seems repetitive. To put it more simply: "How does the initial saturation impact ID thresholds?" However, upon reading the analysis, it seems it would be more accurate to state that the study "evaluates correlation between antecedent saturation and rainfall intensity during landslide events."

P2,L24. Consider providing the inventory with dates here and a table or link to appendix. Plotting landslide locations on the map in Figure 1 would also be great, though perhaps too busy. Figure 1. A scale bar or lat-long coordinates are needed for readers who are not familiar with this region.

P2,L26. Specifically, which meteorological variables are measured? At what timescale?

P3,L3. How are the variables calculated in equation 1? Is the model 1D or distributed in 2D or 3D? What is the spatial and temporal resolution of the model application?

P3,L7. What are the physical parameters?

P4,L17. The statement ". . . a linear decreasing trend between the rainfall thresholds

and initial soil moisture. . ." is somewhat confusing because it is unclear whether Figure 3 intended to illustrate possible thresholds, or merely the decreasing linear trend between rainfall ID and antecedent saturation for landslide events?

Are these colored lines in Figure 3 supposed to be thresholds or merely plots that relate ID to antecedent saturation during landslide events? The y-axis is labelled I/D, but with units of mm/h. It seems that different I/D lines are plotted for uniform duration and different intensities, but the data portion of the plot is unclear. Are the dots data? Are they plotted for a specific timescale/duration? If these are thresholds, they aren't very useful as it seems that about half the landslide events are below the lines.

P4,L18. What is "maximum rainfall" in this context?

P4,L27-28. How did you test different saturation thresholds to arrive at the value of 0.70?

P5,L1. Probability of what? Revise for greater clarity. Table 1 is not clear what the probabilities apply to. What does H1,H2,H3 represent?

Figure 4. Should label that the color bar represents antecedent saturation. I note that for longer duration storms the two thresholds cross, which means that longer storms are more likely to generate landslides when the soil is dry. I cannot imagine a reasonable physical explanation for this, which is problematic.

---

## Short Comment (SC1) · 10 Feb 2019

Dear authors,

I've read your manuscript with interest, both because it concerns a research topic that is relevant to my studies, and because a landslide-rainfall inventory from my native region has been employed to validate the model you proposed.

I must say that I agree with the remarks of the two reviewers about the format of your submission. In fact, I believe more details should be provided to understand and discuss your model, that cannot be contained by a short communication and seem better suited for a full research paper. Alternatively, I may suggest to prepare a substantial supplementary information file, to be attached to the paper, in which all the relevant

details about the landslide-rainfall inventory and about the model can be presented exhaustively.

About the model itself, I believe that the success of I/D thresholds resides in their simplicity and in their empirical nature, so that large amounts of data are readily available for calibrating them, and monitoring data can be used straightforwardly in near real-time early warning systems. On the other hand, I/D thresholds do not say anything, directly, about the actual mechanism of slope failure that leads to the occurrence of landslides of any type.

I believe, and I agree with you in this, that including information on soil moisture obtained from records of antecedent rainfall is one possible strategy to move from fully empirical to at least partly physically-based models, while maintaining the simplicity and immediacy of empirical-only models. On the other hand, I am sure the authors are aware that there is no straightforward connection in most cases between changes of soil moisture and slope instability. Soil moisture is not only markedly variable in space on the slope surface, but might also present significant gradients with depth.

Furthermore, slope instability does not occur, in most cases, as a bulk instability, i.e. as a complete collapse of an entire soil or rock column. Especially in Basilicata, a region characterized by abundant clayey outcrops, the occurrence of this mechanism is probably limited if not absent (while it is typical, for instance, in loess formations, or in loose coarse deposits). On the contrary, slope failure in soil slopes occurs through strain localization at a specific depth (that depends on local geometry, lithology and structure of the slope, and pore water pressures/suctions and soil moisture distribution) which causes the formation of a continuous shear zone or surface, and results eventually in landsliding.

Of course, I understand that the model must be simplified when applied at regional scale, but at the same time I wish that the authors include some discussion about this point: i.e., when you speak of soil moisture, how do you relate it to the hydro-
mechanical condition in the subsurface, where the strain localization that initiates landsliding actually occurs?

One additional remark I wish to make concerns the actual relevance of changes of soil moisture to landsliding. In fact, for landslides in which the shear zone is located even just a few meters deep, and which occur in clay-rich materials (which, again, is a common condition in Basilicata), the variation of soil moisture below the first 1-2 metres from the surface might be small or negligible throughout the hydrological year, and the shear zone might be always fully saturated. In such case, changes of soil moisture above the (potential) shear zone only have a limited effect on the stress state of the shear zone material, by changing the weight of the (potential) landslide body. However, this is only a marginal reason for landsliding, the most important one being the loss of suction or the increase of pore water pressures, that cause a decrease in the effective stress and consequently a decrease in the available resistance to shearing. I think this is a point worth of discussion for the significance and applicability of the model, also in relation to the landslide data set employed for its validation.

---

## Author Comment (AC2) · 26 Mar 2019

Dear Editor, here is a note of reply to the comments of the reviewers. We thank the referees very much for their hard work and constructive criticism. Many observations are worthy of consideration and will allow us to improve and rebalance different parts of the work. At the same time we must point out that being this a short communication, we had to summarize many of our sentences resulting in lack of information deemed necessary by the same referees. Based on your indication, here we don't provide a revised manuscript, but we just reply to ACs and SCs comments by following the suggested structure ((1) comments from Referees, (2) author's response, (3) author's changes in manuscript). We have effected a fusion of point (2) and (3) into one.

[Figure]

Reply to Mirus (Referee 2)

My primary concern when evaluating the technical merit of this work is the lack of details available about the data, modeling, and analysis used to develop the thresholds and arrive at the conclusions. Therefore, the manuscript should undergo major revisions to address the following general question and issues before the work can be evaluated further.

Comment 1. Information is needed about the size of the field area and the actual variables measured.

Reply 1) Please refer to replies 1,2,3 to referee 1

2. Details are missing about the model equations, input/output, physical parameters, parameterization or calibration, and spatial and temporal resolution. The soil water balance has been published elsewhere and the mathematical equation can be found on the manuscript by Manfreda et al. (2017). We consider useless to include all the details of the model in a short communications. The aim of the manuscript is not to validate a physically consistent hydrological model, but it to identify the value of the antecedent soil moisture condition in landslide predictions. Reviewer can argue that as long as the model is not calibrated the result are not valid, but it would be hard to validate a model like this with soil moisture measurements that are not available anywhere in the world. So, the only option available is to reconstruct numerically a reference value of the antecedent soil moisture using all the physical information available. This is exactly what we have done. We may agree on the fact that critical functions should not be considered for operative purposes, because the soil moisture values are not validated. Nevertheless, the value of the research is mainly of the potential of including this information in the actual methodologies rather than in the operative value of the model.

3. The simulated soil moisture does not seem to be compared to any observed soil moisture data, so the accuracy of the model output is highly uncertain. Please refer to

the above comment.

4. It is unclear how the various antecedent saturations were quantitatively evaluated to select the significant value of 0.70. This value was set based on a sensitivity analysis that leaded to the two distinguished sub-samples. We agree with the reviewer that this choice will be better emphasized in the final version of the manuscript.

5. It appears that the thresholds for >0.70 and <0.70 were optimized by identifying a best-fit line through the landslide data, rather than identifying a threshold that distinguishes between landslide and non-landslide events. This is highly unusual and needs to be justified. We agree with the reviewer. The functions reported in the graph are not the threshold functions associated to a given return period and are not based on any optimization process. The functions are just regression linear functions that have been plotted to describe the dependence of triggering rainfall threshold from antecedent soil moisture.

6. Furthermore, there does not appear to be any quantitative analysis of the threshold performance for predicting landslide events, such as ROC analysis or other statistical metrics. We agree also on this point. The manuscript only want to emphasize the role of antecedent soil moisture on the rainfall triggering events. The subsequent step will be the development of a landslide prediction model that will be calibrated considering both landslide and non-landslide events. Any reference to operative methods will be removed from this manuscript for sake of clarity.

Specific comments. The abstract needs to focus more on the actual study and results. The motivation is important, but an informative abstract should include a clear description of the methods and state the primary contributions.

Reply 7) The abstract will be rewritten taking into account both the referees concerns

P1,L11. Preventing landslides seems like a nearly impossible goal and beyond the scope of this study, but reducing the losses and impacts is more achievable through

developing better landslide thresholds for early warning. That is the focus of this work.

Reply 8) We agree with the referee comment and we will adjust the paragraph.

P2,L1&4. The Mirus et al. 2018a,b references might be more appropriate to cite in line1 as we actually developed new thresholds that improve predictive capabilities, though they do not explicitly consider rainfall I/D. Also consider adding Thomas et al., 2018, Geophysical Research Letters, doi:10.1029/2018GL079662, which uses a deterministic approach with infiltration simulations to identify rainfall-saturation thresholds. P2,L8-9. Thomas et al., (2018) also does a nice job of quantifying the sensitivity of thresholds for different hydraulic and strength properties in relation to rainfall-saturation thresholds, though it doesn't deal with I/D directly. Godt et al., 2006 directly examines how antecedent moisture index affects the accuracy of an I/D threshold. It is not clear how/why your objectives should be distinguished from these prior advances.

Reply 9) We will rewrite the paragraphs by following the referee suggestions and by also considering the new insights of Thomas et al., 2018. This last paper will be cited in the Reference list

P2,L9. Question seems repetitive. To put it more simply: "How does the initial saturation impact ID thresholds?" However, upon reading the analysis, it seems it would be more accurate to state that the study "evaluates correlation between antecedent saturation and rainfall intensity during landslide events."

Reply 10) The statement will be rewritten

P2,L24. Consider providing the inventory with dates here and a table or link to appendix. Plotting landslide locations on the map in Figure 1 would also be great, though perhaps too busy. Figure 1. A scale bar or lat-long coordinates are needed for readers who are not familiar with this region.

Reply 11) The Figure 1 has been redrawn by taking into account both the referees comments

P2,L26. Specifically, which meteorological variables are measured? At what timescale? P3,L3. How are the variables calculated in equation 1? Is the model 1D or distributed in 2D or 3D? What is the spatial and temporal resolution of the model application? P3,L7. What are the physical parameters?

Reply 12) Some additional information about the hydrological model will be included in the revised version of the manuscript. Just for clarity, the model is a conceptual model applied at the catchment scale, its parameters are associated to the soil texture, and it is applied at the daily scale.

P4,L17. The statement ". . . a linear decreasing trend between the rainfall thresholds and initial soil moisture. . ." is somewhat confusing because it is unclear whether Figure 3 intended to illustrate possible thresholds, or merely the decreasing linear trend between rainfall ID and antecedent saturation for landslide events? Are these colored lines in Figure 3 supposed to be thresholds or merely plots that relate ID to antecedent saturation during landslide events? The y-axis is labelled I/D, but with units of mm/h. It seems that different I/D lines are plotted for uniform duration and different intensities, but the data portion of the plot is unclear. Are the dots data? Are they plotted for a specific timescale/duration? If these are thresholds, they aren't very useful as it seems that about half the landslide events are below the lines.

Reply 13) The objective of Figure 3 is to show the behaviour of rainfall thresholds in function of initial degree saturation and the event duration. The colored lines plot the relation between ID and antecedent saturation during landslide event. Thus, as the referee marked, the units of mm/h is wrong (it is mm).

P4,L18. What is "maximum rainfall" in this context?

Reply 14) It is an error. It simply refers to rainfall

P4,L27-28. How did you test different saturation thresholds to arrive at the value of 0.70?

[Figure]

Reply 15) We performed a sensitivity analysis in order to identify the threshold able the better distinguish between the two groups of events. This point will be better discussed in the manuscript.

P5,L1. Probability of what? Revise for greater clarity. Table 1 is not clear what the probabilities apply to. What does H1,H2,H3 represent?

Reply 16) Please refer to reply 5 to referee 1

Figure 4. Should label that the color bar represents antecedent saturation. I note that for longer duration storms the two thresholds cross, which means that longer storms are more likely to generate landslides when the soil is dry. I cannot imagine a reasonable physical explanation for this, which is problematic.

Reply 17) The colorbar represents the relative soil saturation ranging from 0 to 1. The two functions plotted in the graph have significantly different slopes. This implies that they must cross somewhere in the space of rainfall intensities and event duration. In the present case, we observe that they cross in a point corresponding to duration of about 200h. At such duration, the impact of antecedent soil water content becomes not relevant and this part of the curve should not been considered. This is a very good point that will be also mentioned in the discussion of the manuscript.

---

## Author Comment (AC3) · 26 Mar 2019

Comment 1_I've read your manuscript with interest, both because it concerns a research topic that is relevant to my studies, and because a landslide-rainfall inventory from my native region has been employed to validate the model you proposed. I must say that I agree with the remarks of the two reviewers about the format of your submission. In fact, I believe more details should be provided to understand and discuss your model, that cannot be contained by a short communication and seem better suited for a full research paper. Alternatively, I may suggest to prepare a substantial supplementary information file, to be attached to the paper, in which all the relevant details about the landslide-rainfall inventory and about the model can be presented exhaustively.

[Figure]

Replay 1_We would like to express our gratitude for the manifestation of interest and the comments provided. We agree with you and following your suggestion and also the requests of the referees, the manuscript will be reorganized keeping the format of short communication, but integrating with supplementary material the manuscript. In particular, we decided to provide the full database of the landslide events considered.

Comment 2_About the model itself, I believe that the success of I/D thresholds resides in their simplicity and in their empirical nature, so that large amounts of data are readily available for calibrating them, and monitoring data can be used straightforwardly in near real-time early warning systems. On the other hand, I/D thresholds do not say anything, directly, about the actual mechanism of slope failure that leads to the occurrence of landslides of any type. I believe, and I agree with you in this, that including information on soil moisture obtained from records of antecedent rainfall is one possible strategy to move from fully empirical to at least partly physically-based models, while maintaining the simplicity and immediacy of empirical-only models. On the other hand, I am sure the authors are aware that there is no straightforward connection in most cases between changes of soil moisture and slope instability. Soil moisture is not only markedly variable in space on the slope surface, but might also present significant gradients with depth.

Replay 2_The main objective of this paper is to define the possible role of soil moisture and saturation degree in the shallow landslides triggering, whose sliding surface is defined in the first few meters of depth, always bearing in mind, however, that the soil moisture is not the only factor that determines the evolution of the landslides but, in many cases it can be decisive as well as the lithological conditions, slope and aspect, vegetation coverage and the presence of a water table. The model proposed on a regional scale considers homogeneous soil moisture conditions in the space and in the first meters of depth in the areas affected by each landslide considered in the our database. Therefore, the considered antecedent soil moisture must not be confused with the specific distribution of soil water content in the slope, but it is representative of

the mean antecedent conditions that are neglected otherwise.

Comment_3 Of course, I understand that the model must be simplified when applied at regional scale, but at the same time I wish that the authors include some discussion about this point: i.e., when you speak of soil moisture, how do you relate it to the hydro-mechanical condition in the subsurface, where the strain localization that initiates landsliding actually occurs?

Replay 3_ The methodological approach used provides an assessment of the relationship between soil moisture and landslides on a regional scale, without considering the specific site conditions. A forthcoming extension of this research will aim to carry out a local downscaling to define the relations between I / D and the degree of soil saturation in the smallest territorial contexts characterized by the same climatic and lithotechnical conditions, in which the landslides inserted in our database have developed.

Comment_4 One additional remark I wish to make concerns the actual relevance of changes of soil moisture to landsliding. In fact, for landslides in which the shear zone is located even just a few meters deep, and which occur in clay-rich materials (which, again, is a common condition in Basilicata), the variation of soil moisture below the first 1-2 metres from the surface might be small or negligible throughout the hydrological year, and the shear zone might be always fully saturated. In such case, changes of soil moisture above the (potential) shear zone only have a limited effect on the stress state of the shear zone material, by changing the weight of the (potential) landslide body. However, this is only a marginal reason for landsliding, the most important one being the loss of suction or the increase of pore water pressures, that cause a decrease in the effective stress and consequently a decrease in the available resistance to shearing. I think this is a point worth of discussion for the significance and applicability of the model, also in relation to the landslide data set employed for its validation.

Replay 4_The mentioned processes is likely to occur and there may be cases were the inclusion of antecedent soil moisture condition do not provide any help in the description of the process. Our considerations are made at the regional scale and this may mask specific processes such as the one mentioned above. In a conceptual scheme, it would be hard to include this mechanism of predict them without a
* * *